# The LED Paradox: How Light Pollution Challenges Experts to Reconsider Sustainable Lighting

**Nona Schulte-Römer [1],\* , Josiane Meier [2], Max Söding [2] and Etta Dannemann [3]**

1   Helmholtz Centre for Environmental Research—UFZ, 04318 Leipzig, Germany
2   School of Planning-Building-Environment, Technische Universität Berlin, 10623 Berlin, Germany;
    josiane.meier@tu-berlin.de (J.M.); m.soeding@campus.tu-berlin.de (M.S.)
3   Studio Dannemann, 12435 Berlin, Germany; etta@studiodannemann.berlin
\*   Correspondence: nona.schulte-roemer@ufz.de

**Abstract:** In the 21st century, the notion of "sustainable lighting" is closely associated with LED technology. In the past ten years, municipalities and private light users worldwide have installed light-emitting diodes in urban spaces and public streets to save energy. Yet an increasing body of interdisciplinary research suggests that supposedly sustainable LED installations are in fact unsustainable, because they increase light pollution. Paradoxically, blue-rich cool-white LED lighting, which is the most energy-efficient, also appears to be the most ecologically unfriendly. Biologists, physicians and ecologists warn that blue-rich LED light disturbs the circadian day-and-night rhythm of living organisms, including humans, with potential negative health effects on individual species and whole ecosystems. Can the paradox be solved? This paper explores this question based on our transdisciplinary research project Light Pollution—A Global Discussion. It reveals how light pollution experts and lighting professionals see the challenges and potential of LED lighting from their different viewpoints. This expert feedback shows that "sustainable LED lighting" goes far beyond energy efficiency as it raises complex design issues that imply stakeholder negotiation. It also suggests that the LED paradox may be solved in context, but hardly in principle.

**Keywords:** sustainable lighting; light-emitting diodes (LEDs); innovation; artificial light at night (ALAN); outdoor lighting; light pollution

## 1. Introduction

"LED lights are the best eco-friendly lighting options available" announced Light + Building, the world's leading trade fair for lighting, in an online post in August 2019. Indeed, the energy savings using light-emitting diodes (LEDs), also known as solid-state lighting (SSL), are compelling. In 2011, the European Commission's Vice President Neelie Kroes argued in a press release: "Expanding LED lighting is a 'no-brainer'. It means more money in your pocket, and a healthier planet." [1]. The European Commission further expects that LED lighting can "make our cities 'greener' by saving up to 70% of lighting energy and reducing costs compared to existing lighting infrastructures." [2] (p. 5). Similarly, the US Department of Energy reports on its website that municipalities that have switched to new LED technologies could "reduce energy costs by approximately 50% over conventional lighting technologies." [3]. In India and China, governments have launched large-scale programs for replacing conventional streetlights with energy-efficient LEDs [4,5]. Yet, environmentalists are increasingly alarmed. Although they are in favor of energy savings, they object to and even campaign against the implementation of LED technology claiming that artificial light at night (ALAN) has adverse effects on nocturnal ecosystems. In 2003, Travis Longcore and Catherine Rich coined the term "ecological light pollution" to highlight these effects [6]. Cool-white LEDs, which are particularly

energy efficient, are also considered particularly problematic due to their blue-rich light spectrum. In 2016, the American Medical Association (AMA) warned municipalities against the potential negative side effects and health impacts of outdoor LED lighting [7]. The doctors echo biologists' concerns that light at the wrong time—and especially intense blue-rich LED light at night—can disrupt the natural day–night rhythms of wildlife, plants and entire ecosystems [8,9].

The controversial expert views reveal a paradox: The implementation of supposedly sustainable, energy-efficient LED technology for climate change mitigation can have very unsustainable, ecologically unfriendly side effects on flora, fauna and humans, and ultimately on biodiversity. While the worldwide diffusion of LED technology is still ongoing and supported by innovation programs, environmental concerns create uncertainty among light users regarding the sustainable appeal of the new technology as an important selling argument. Governments invest in the study of adverse ecological and health effects caused by LED lighting [10–12], while continuing to support its deployment as part of their climate change mitigation and innovation policies. As public complaints about light pollution become louder, LED lighting is no longer a "no-brainer", but rather an incalculable risk.

From an analytical point of view, emerging paradoxes offer both scholars and policymakers the chance to recognize incompatibilities and inconsistencies in their theories and practice. "In general parlance", explain Poole and Van de Ven, the term is used "as an informal umbrella for interesting and thought-provoking contradictions. . . . In this sense, a paradox is something which grabs our attention, a puzzle needing a solution" [13] (p. 563). The recognition of a paradox thus offers the "possibility of becoming knowledgeable about one's own ignorance" [14] (p. 23). Accordingly, the paradox of unsustainable "sustainable" LED lighting has drawn attention to the multifaceted nature of artificial light at night, which goes far beyond energy efficiency. Or to reframe it as a research question: What can we learn from the paradoxical situation about sustainable LED lighting?

In this article, we explore the LED paradox through the lens of experts with a focus on the issue of light pollution. The experts all participated in our transdisciplinary research project Light Pollution—A Global Discussion, and they view the problem from diverse perspectives. While some qualify as lighting experts because they make light professionally as lighting designers, light planners and manufacturers, others have become experts for light pollution as they care about darkness and study the adverse environmental and health effects of artificial light at night (ALAN) as astronomers, biologists, environmentalists and concerned citizens. Based on the feedback of both expert groups we ask: How can LED technology contribute to sustainable lighting by reducing light pollution? The expert answers show that there are ways to solve the LED paradox and use LEDs in ways that meet economic, social as well as environmental demands [15]. Yet, we also see that these solutions go beyond technological fixes as they challenge light users to rethink their lighting practices. These findings are in line with social-scientific research that shows that technological innovation goes beyond the implementation of new products and processes and depends on producers' and users' capability to deal with novelty and apply it in appropriate ways [16,17].

The paper is structured as follows: In the next section, we outline our approach, data collection and analysis. We then present expert feedback on the LED dilemma with special attention to the positions and views of lighting professionals and light pollution experts. In our discussion, we reflect on these findings with regard to existing research on light pollution and through the lens of social-scientific research on technology and innovation. We conclude by highlighting the social dimension and contexts of socio-technical innovation and their implications for sustainable lighting design.

## 2. A Transdisciplinary Approach: Light Pollution—A Global Discussion

The results presented in this paper are part of a greater, transdisciplinary research project called Light Pollution—A Global Discussion [18]. We developed the project based on our complementary analytical lenses and expertise in different parts of the lighting field [19–21]. A key concern was to collect expert views from all over the world and present the full breadth and depth of those voices. The project first went public in March 2018 in the form of an international online survey. The survey

addressed respondents with expertise in lighting, including lighting professionals, scientists, dark sky activists, hobby astronomers and conservationists, to gain an overview over debates on light as a form of pollution. It was circulated via social media channels, personal emails and mailing lists (snowball principle) in professional circles of lighting designers, light planners and others in the lighting industry, as well as the emerging community of researchers on artificial light at night (ALAN) and dark-sky movements. Over 250 respondents from 36 countries with a wide range of lighting-related professional backgrounds shared their experiences and opinions on light pollution and best practice in lighting (205 completed the survey). The respondents were aged between 20 and 79, and about one third was female. Most of them (101; 65%) were based in Europe, while 29 (19%) were based in Anglo America, 13 (8%) in Australia and New Zealand, 5 (3%) in Middle Eastern or African countries, 4 (3%) in Latin America and 3 (2%) in Asia [22] (see also the supplementary material: www.ufz.de/light-pollution). The questionnaire included closed questions (Likert scales, multiple choice and single choice questions) as well as open questions, where respondents could spell out their views in more detail and in their own words.

The expert feedback offers rich material for analyzing the LED paradox, or more precisely, the relationship between energy-efficient and eco-friendly lighting. Focusing on the open expert statements, we categorized the arguments for and against LED lighting in the course of an iterative and inductive coding progress and compared the categories in a qualitative frequency analysis [23]. To explore the results in more detail, we identified different groups of survey participants based on their professional and non-professional activities around lighting. These groups formed the basis for our differentiation between "lighting professionals" and "light pollution experts". The group of lighting professionals (*n* = 67) includes all those respondents who professionally plan, design, or produce artificial light and lighting technology. The light pollution experts (*n* = 89) are mostly astronomers, conservationists, natural and social scientists who problematize artificial light at night (ALAN) from their various viewpoints. The categorization represents ideal types. In reality, the two groups can overlap. Some lighting professionals actively engage in raising awareness for light pollution and developing solutions for its mitigation. Others have acquired detailed knowledge of lighting technology and lighting practices and advise municipalities on sustainable lighting [22] (p. 3). In our quantitative analysis of the survey data, we juxtaposed the answers of the two groups. For the quantitative analysis, we used the statistical software R [24]. Figures were produced with Microsoft Excel software.

While the expert survey revealed the bigger picture, three focused group interviews allowed us to contextualize the survey results in relation to the lighting field and societal demands for light and darkness [18] (pp. 31–168). Again, we made sure that the expert interviewees in each group discussion had different professions and came from different parts of the world. Before the curated discussions, they received introductory information about the main focus of the discussion and our core questions. In June 2018, each of the three conference calls brought together four experts from different continents to discuss controversial issues around light pollution, namely the protection of dark skies, the question of color temperature in street lighting, and how to regulate private lighting in cities. Taken together, our transdisciplinary approach and the non-representative survey offer rich data for analyzing the LED paradox from a broad international perspective and in practical detail.

## 3. Results: Ambivalent Views on LED Lighting

The survey feedback shows that the expert participants in our transdisciplinary project are well aware of the unsustainable side effects of LED technology. Both lighting professionals and light pollution experts see a strong relationship between LED lighting and environmentally unfriendly artificial light at night. Lighting designer Paulina Villalobos from Chile worries in one of our expert discussions: "There is like an avalanche of change towards the worst possible type of LED. If they keep on changing at this rate, [ . . . ] we are going to erase all nocturnal life" [18] (p. 52). The majority of the survey participants raised similar environmental concerns. About 80% found that a general lack

of awareness and the introduction of LEDs without consideration of negative side effects was a major obstacle to avoiding light pollution [18] (pp. 193–194).

Negative side effects associated with light pollution include negative impacts on wildlife, ecosystems and people's sleep. In the survey, these were also the highest-ranking arguments for reducing light pollution with over 85% consent among all participants, both light pollution experts and lighting professionals [18] (pp. 181–182).

With regard to light pollution, the experts see the sustainability potential of LED technology with ambivalence. This is most obvious in the responses to the multiple-choice question regarding the relationship between LED lighting and light pollution (Figure 1): 70% of the survey participants find that LEDs contribute to light pollution, while 45% think the innovation helps to reduce light pollution. The overlap shows that one third of the respondents could agree with both statements.

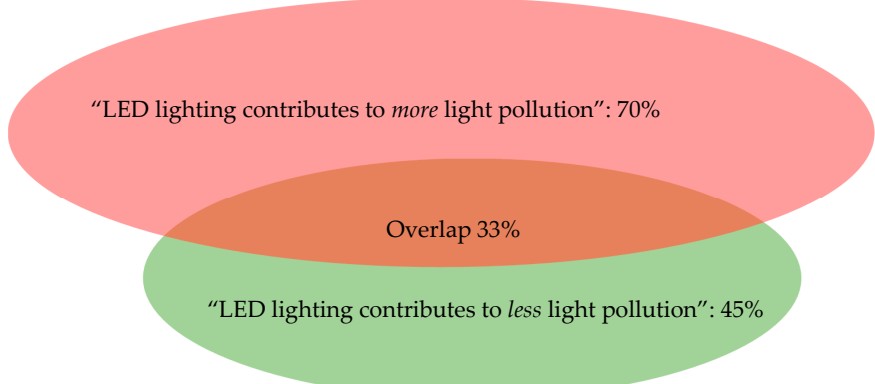

**Figure 1.** Responses to the multiple-choice question "How do you see the relationship between LED lighting and light pollution?" (all survey respondents, *n* = 214).

The ambivalence remains, even when the supposedly more critical answers of light pollution experts (*n* = 89) are separated from the supposedly less critical views of lighting professionals (*n* = 67) in our sample. Among the light pollution experts, 90% (80) are critical of LEDs, whereas 43% (38) of the respondents in this group support the claim that LEDs can reduce light pollution (Figure 2). Almost as many of them (36) are ambivalent towards LED lighting and agreed to statements both for and against LEDs. Among the lighting professionals, 42% (28) think that LEDs reduce light pollution, whereas a surprisingly larger share of 49% (33) responded that they cause light pollution. The overlap in the group of lighting professionals is only 19%.

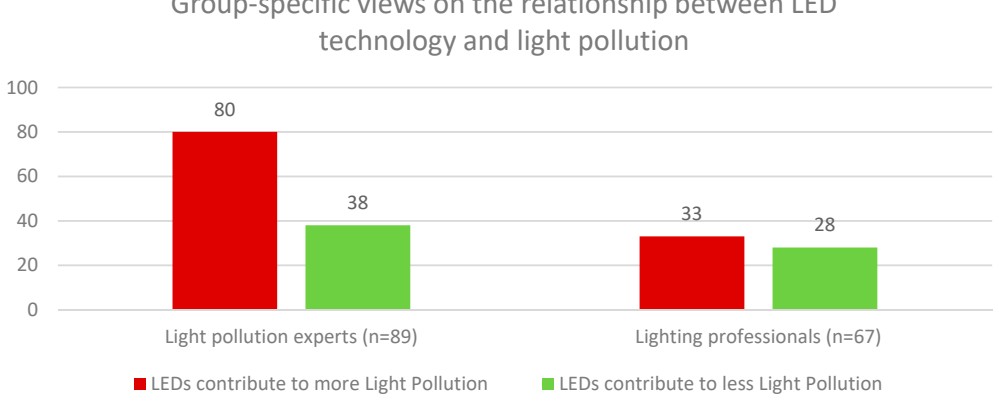

**Figure 2.** In the group of light pollution experts, 36 respondents argued against and in favor of LED technology; in the group of lighting professionals, 13 respondents argued against and in favor.

Many of the respondents further explained their views on LED lighting and light pollution in open statements. In the following two sections, we summarize their statements against and in favor of LED lighting.

### 3.1. *"Cheaper, More Blue and Brighter": LEDs Contribute to Light Pollution*

In total, 137 light pollution experts and lighting professionals explained why LEDs contribute to more light pollution and 92 explained why they can reduce light pollution. When asked why they thought that LED technology contributed to more light pollution, they mentioned LED-related problems and usage-related problems that are well known in expert communities but have not yet been solved. LEDs add to light pollution since they are "cheaper, more blue and brighter", summarized one light pollution expert very pointedly. Other statements offered slightly more detail, but were generally short and concise. In our qualitative analysis, we categorized them into six different arguments plus one residual category. Table 1 lists the categories in order of their importance and illustrates them with exemplary survey statements.

It is obvious that the arguments operate on different levels and are not mutually exclusive. Some respondents address several LED-related physical aspects like light levels, glare and light spectrum in one sentence. Others explain why these physical aspects matter in the context of light pollution, for instance, by pointing to circadian disruption. Usage-related statements show that the physical aspects of LED lighting are not set in stone, but result from design choices and specific usages.

**Table 1.** Why LED lighting contributes to more light pollution—open statements from the survey.

| Code Categories | Exemplary Statements |
|---|---|
| Blue-rich color temperature (54.3%) | "[because] most LED have a large blue light output." <br> "blue-rich light causes more pollution and has a more deleterious effect on wildlife and the environment" <br> "Increased Rayleigh scattering from short-wave light." <br> "Blue White light, and no thought of narrow band amber bulbs" |
| Rebound effects (43.8%) | "More fixtures may be installed, due to higher efficiency and lower connected load." <br> "Due to sinking costs for maintenance more fixtures are installed (rebound effect)" <br> "… overuse of color 'because we can'" <br> "Because of the so-called rebound effects and the many possible applications of LEDs." |
| Increase in brightness (23.8%) | "… because of misunderstanding about what lighting levels can be achieved." <br> "Due to higher lumen/Watt ratios (and unfortunate standards … )" <br> "… much brighter and glary light compared to traditional lighting sources." |
| Bad design choices, bad installations (13.3%) | "… communities do not care about the real possibilities by dimming them [LEDs]." <br> "Many local authorities use over-bright, too-blue LEDs. Most domestic and commercial exterior lighting is poorly designed, too bright and has little or no information about preventing pollution." <br> "… LEDs increase light pollution when they are used as replacements for CFL lamps because their higher efficiency is not used to reduce the energy consumption but to increase the level of lightning." |
| Glare (9.5%) | "… much brighter and glary light compared to traditional lighting sources." <br> "… intense distribution of light/glare" <br> "It creates … debilitating glare … " <br> "Strong directional project contributes to glare, and can require more installations." |
| Poor luminaire design (7.6%) | "… the only LED streetlight available here in Brazil so far is 6.400k!" <br> "Illumination levels too high; retrofit are bad and still many unshielded LED will be established." <br> "Horrible 'white' spectrums. Planar wavefront formation, concentrated source surfaces … " |
| Other (7.6%) | "It creates more circadian disruption, debilitating glare and sky glow + is an aesthetic disaster." <br> "LEDs terrorize my eyes virtually everywhere - in streetlamps, at sports fields, in DRLs, in ceilings, even in some new refrigerators (which have two strips of LEDs)." <br> "NASA photos prove there is more light pollution. My burning retinas also register more light." |

When comparing the arguments of light pollution experts and lighting professionals, we find interesting commonalities and differences (Figure 3). Among the light pollution experts, almost two thirds (65%) highlight the problem of color temperature; that is, the blue-rich spectrum of LED lighting. Just over one third (36%) mention the issue of rebound effects. More than one in four light pollution experts (27%) point to generally brighter light levels and about 9% complain about glare, often combined with remarks about poor, unshielded luminaire designs (4%) and bad lighting designs and installations (12%). In comparison, the lighting professionals in our sample mostly highlight rebound

effects (63%), followed by issues with blue-rich color temperatures (27%), increasing light levels, poor luminaire and lighting design and/or bad installations (all 17%), and glare (10%).

The differences between the two groups seem related to the different foci of the two groups. Lighting professionals experience the increasing use of LED lights, their clients' demands, unsatisfying LED products and bad lighting design and planning on a daily basis, which can explain their greater awareness of these issues. This became particularly clear in our thematic discussion about the regulation of city lights, where lighting designer Cinzia Ferrara expressed her concern about "a world that doesn't have any limits", where people always desire more [18] (p. 165). Moreover, many of the lighting professionals consider excessively bright light levels and glare to be effects of poor design and hence, a result of problematic planning and installations. In line with this feedback, Allan Howard, another discussion participant who works as a lighting and building consultant for an international engineering company, stressed the importance of professional lighting expertise in building projects, which also means ensuring "that the contractor doesn't substitute the (lighting) designers" [18] (p. 153).

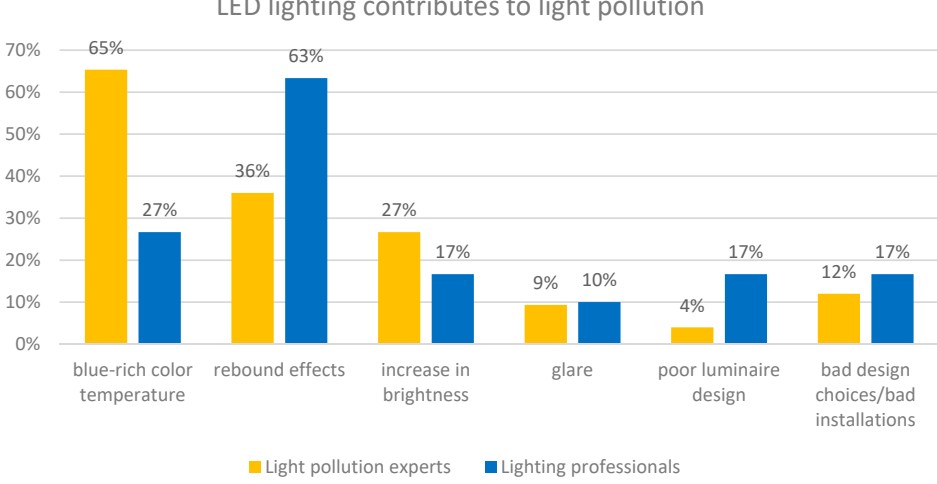

**Figure 3.** Comparison of LED-critical open statements by category and group (*n* = 105, multiple statements possible).

In contrast, light pollution experts are more concerned about color temperature. This is not surprising, considering that the light spectrum plays a key role in ecological, medical, as well as astronomical research on artificial light at night [25]. Astronomers know that blue-rich lighting scatters more in the atmosphere (Rayleigh scattering) and thus contributes to undesirable sky glow, which is also reflected in the scientific literature [26–28]. Neuro-scientists and biologists show that blue-rich light at night has the greatest effects on the circadian rhythm of humans [29–31] and on a number of wildlife species [9,32–34]. They further recommend light sources with narrow spectral ranges, as ecologist Sybille Schroer explained in our thematic discussion on color temperature choices: "[T]he broader the spectrum . . . , the broader the group of organisms you affect. Because light is never neutral, it affects organisms in different ways and some are more sensitive to short wavelengths, others to longer wavelengths" [18] (p. 80).

In our survey, the number of lighting professionals who found blue-rich LED light problematic was much smaller (27%) than the number of light pollution experts (65%). The thematic discussion on color temperature revealed that there is a deep-seated, underlying conflict that cannot be easily solved. As opposed to the eco-friendly solution described above, lighting professionals prefer full-spectrum light sources and white light, which includes the blue part of the spectrum. The reason for this choice is twofold, as lighting engineer Tran-Quoc Khanh pointed out: "With blue-rich white light, we have both—more brightness and a better contrast on the street with the same amount of wattage. This means that we could reduce our wattages and energy use in cities and get the same amount of visibility that we have today" [18] (p. 87). This is because the human eye is best adapted to full-spectrum daylight.

Under twilight conditions, it is even more sensitive to shorter wavelengths [35]. In addition to this physiological advantage, there is also a positive physical effect. Cooler LEDs are more energy efficient. Since the color conversion of blue LED light into white light via phosphor coatings absorbs energy, the energy loss is smaller in blue-rich LEDs.

In line with these photometric findings, lighting designer Nancy Clanton shared the results of a field experiment where people drove a car at 35 miles per hour (50 km/hour) and pressed a button as soon as they noticed an object on the street. Clanton reports that the 4000 Kelvin color temperature "significantly increased the reaction time" and "white LED light and a better color rendering helps to increase the visual performance of drivers compared to high pressure sodium" [18] (p. 90). At the same time, Clanton and her discussion partners from India and Germany observed that there are places where citizens object to cool-white street lighting. Venkatsh Dwivedi, who is overseeing the large-scale LED refurbishment of public streetlights in India [4], reported that they chose cool-white LEDs (5000 and 5700 Kelvin) "for efficiency reasons and . . . because the color looked much better". Yet, when they changed the lighting around iconic places and heritage sites, citizens complained until a court decided that the light should be made warmer [18] (p. 97). Citizen protests against cool LED light have also taken place in other places like Montreal, New York, Rome and Berlin [18] (pp. 185–186) [36].

Coming back to the survey results, half of the respondents mention not just one, but two to four aspects in their short statements. For instance, one in six respondents problematizes the blue-rich color temperature of LEDs and rebound effects in one breath: " . . . [because] most LED have a large blue light output and are used more excessively [because] of reduced energy cost." Similarly, almost one fifth of the light pollution experts refer to color temperature in combination with increased brightness. Yet, while these physical aspects might contribute to light pollution independently of one another, other statements point to causes and effects and show clearly that physical LED-related aspects like blue-rich color temperature, glare and excessive light levels are often inseparable from (if not the result of) usage-related design choices.

Such problematic aspects arise during all stages of the innovation process, starting with the development of new LED products, the choice and procurement of new luminaires, the infrastructure planning and the installation on site. Experts are well aware of the pitfalls of professional lighting practices. As one light pollution expert points out, "lighting installations are never tested or checked at night." Another suggests that light pollution increases because of "bad retrofits" and the existence of "many unshielded LEDs." Retrofits are causally linked to higher light levels "when people do one-for-one swaps without considering higher visibility under LEDs, which could allow for overall lower light levels."

Meanwhile, light pollution experts also criticize the fact that municipal LED users "do not care about the real possibilities (of LED technology) by dimming them." In the same vein, a lighting professional criticizes "excessive spectrum at damaging wavelengths and poor glare control." Another points out: "Poorly designed installations, and especially those that do not dim LED installations (relative to earlier technologies), are shown to result in greater glare and sky glow." Yet another addresses project-level design decisions, market-level LED product designs and worldwide rebound effects in one sentence: "Horrible 'white' spectrums. Planar wavefront formation, concentrated source surfaces. High efficiency leads to over-illumination." The comment also illustrates that rebound effects can be both quantitative (more light points) and qualitative (brighter light points).

Last but not least, several respondents criticize a general lack of consideration in the lighting field: "There is no holistic design intention", complains a lighting professional, while others complain about the "specifiers" who bring LED technology into the real world through their planning and product choices. One lighting professional says specifiers do not do enough research and adds "cheapest is not best". Similarly, another argues that "specifiers consider lumens per dollar, but seem to go no further on actual outcome or effect." When considering poor design as part of the problem, it seems that unsustainable LED lighting does not result from the technology itself, but rather from

unsustainable usage and design practices that do not adequately capitalize on the novel characteristics of the innovation. This impression is reinforced by our analysis of the arguments that suggest LED lighting can reduce light pollution.

### 3.2. *"If Correct CCT and Good Design": LEDs Reduce Light Pollution*

For many respondents from both groups, and in line with the previous findings, the innovation's potential is contingent on "correct CCT and good design". For example, one expert states: "LEDs can be shielded to reduce light pollution. They can be switched on by demand and have several control options. They come in low CCTs for healthier light." CCT stands for correlated color temperature, which can be adapted and even dynamically controlled in LED luminaires. "Good design" can thus refer to spectrum control, but also light levels, light distribution and even dynamic control.

To look more closely at why the survey participants thought LED lighting can reduce light pollution, we identified four arguments plus one residual category. Table 2 offers an overview of these arguments in order of importance, with exemplary statements from the survey.

**Table 2.** Why LED lighting contributes to less light pollution—open statements from the survey.

| Code Categories | Exemplary Statements |
|---|---|
| Good luminaire designs and installations (inseparable) (82.8%) | "Because LEDs are more flexible than other light sources and can bring light only to areas where it is really needed." <br> "LEDs are key to reducing light pollution: directionality, dimmable, tailored spectrum." <br> "LEDs can be shielded to reduce light pollution … " <br> "Better inherent shielding directional control in many designs." |
| Low color temperature (40.6%) | "If having a color temperature < 2700 K. There are orange LEDs with a TCC < 2200 K." <br> "The option of low-blue-light LED like PC-amber can offer a great combination of the pros of LED while reducing the amount of blue to amounts that are less than HPS." <br> " … they come in low CCTs for healthier light." <br> " … PC amber LEDs may give a substantial reduction of LP." |
| Dimmed light levels or temporal darkness (37.5%) | "It can be dimmed easily, it can be easily directed because of the optical possibilities, and the spectrum light emissions can be regulated and customized." <br> "Dimming and shutoff options could help reduce light pollution by shutting off lights when not needed." <br> "If used correctly, LEDs can reduce light pollution. It all depends on the light distribution and of course dimming." |
| Dynamic digital and/or sensor-based control (25%) | " … better options for intelligent dimming/switching (sadly not utilized in many areas)" <br> "Smart lighting allows dimming when required" <br> "LEDs enable the development and the use of smart lighting solutions (dimming, instant detection … )" |
| Other (9.3%) | "Proper binning, right CCT, adequate testing and proper disposal." <br> "Higher [product] prices allow better beam shaping" |

This time, we do not distinguish between luminaire and lighting design as they are not clearly discernable in the statements. For instance, the frequently and positively mentioned directed LED light can be the result of a well-shielded luminaire with customized optics and/or the outcome of good light planning and design. We do differentiate between "dimmed light levels" and "dynamic control", even though the aim of sensor-controlled installations is to dim lights or allow temporal darkness—an approach some respondents referred to as "light when needed". Technologically speaking, there is a difference between just dimming or switching off lights and dynamically controlled lights, since the latter requires some form of advanced, dynamic or responsive control system that is either sensor-based or digitally programmed. What is known as "smart" or "intelligent" lighting therefore marks the next step in terms of innovation. Smart lighting is more radical than retrofitting existing infrastructures with LEDs, even if such retrofits can also be dimmed and switched off with non-dynamic technology [21]. This difference is also reflected by the fact that most of the survey

respondents feel affected by LED lighting, while the introduction of dynamic, digitally controlled lighting systems plays a less important role in their light-related activities (the item ranked sixth in a list of ten potential trends, see Appendix A).

In contrast to the arguments against LEDs, the arguments in favor of LEDs are quite homogeneously distributed in the two groups (Figure 4). The majority of light pollution experts (84%) and lighting professionals (81%) argue that well-considered and appropriate LED fixtures and lighting designs can reduce light pollution. A light pollution expert acknowledges that there are generally "more standardly shielded fixtures" and that it is "quite easily possible to reduce the brightness." Several lighting professionals highlight that beam control, meaning controlled light distribution, is easier with laser-like LED light sources: "LEDs are more flexible than other light sources and can bring light only to areas where it is really needed." Another explains in more detail that better beam control means "less uplight and light trespass" and adds that LEDs also show "better dimming capabilities and better spectral control for ecologically sensitive areas (where spectral sensitivity is established, e.g., turtles, migratory birds)", which is a color temperature argument.

In both groups, 41% of the respondents find that the customizable color temperature or "spectral control" of LED luminaires provides an opportunity to reduce the spectral range and make artificial light at night more ecologically friendly and healthy. Some light pollution experts explicitly promote the use of phosphor-coated amber colored LEDs: "The option of low-blue-light LEDs like PC-amber can offer a great combination of the pros of LEDs while reducing the amount of blue to amounts that are less than HPS." On the issue of increasing brightness, 41% of the light pollution experts and 33% of the lighting professionals point out that LED technology facilitates dimming and better light distribution. A lighting professional explains "it can be dimmed easily, it can be easily directed because of the optical possibilities, and the spectrum light emissions can be regulated and customized." Interestingly, only 25% of the survey participants mobilize the marketing argument that LED technology is perfectly suited for "smart" or "intelligent" sensor-controlled lighting systems (Figure 4).

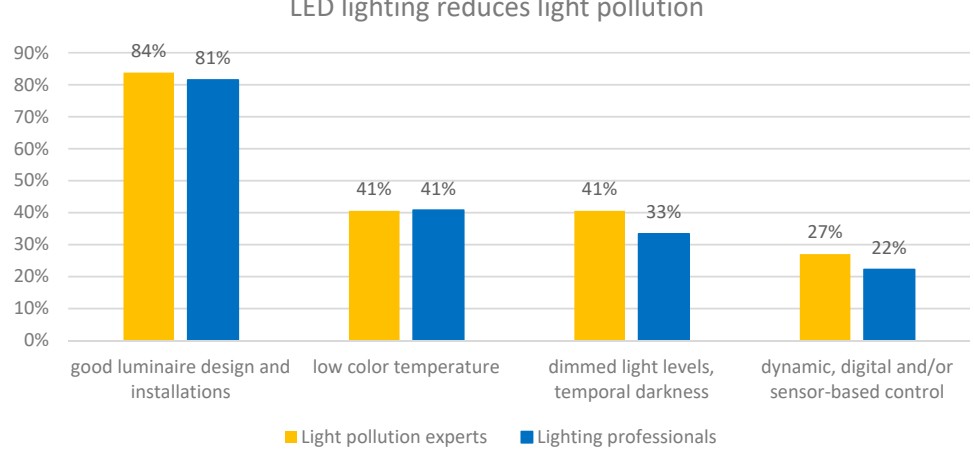

**Figure 4.** Comparison of LED-favorable open statements by category and group (*n* = 64, multiple statements possible).

As with the pro LED statements, lighting design is causally linked to technological facts, as outlined by a light pollution expert: "Well-designed installations, in which light spill is well controlled, CCT is low and illuminance levels are appropriate, could well have the effect of reducing light pollution worldwide." Another describes a trade-off but is still in favor of LEDs: "A recent change from old HPS lighting to LED street lighting showed an increase in sky quality. Although the penetration of higher color temperatures is greater, the directed output meant better results for dark skies."

After having read very strong arguments against LED lighting it might be surprising that the same light pollution experts now highlight the great advantages of LED light: "LEDs are key to reducing

light pollution: directionality, dimmable, tailored spectrum." On the other hand, the expert statements make it clear that the advantages of the innovation depend on how it is used. The experts not only link the potential reduction of light pollution to design choices (>80%), but also stress the conditionality of their LED appraisal in the language they use.

More than half the statements in favor of LEDs contain an "if", "could" or some other conditional grammatical form. To give just a few examples, one lighting professional argues: "LED lighting is a big chance to use the appropriate amount, the perfect light and spectral distribution for the intended purpose." Another explains: "If used correctly, LEDs can reduce light pollution. It all depends on the light distribution and of course dimming." Some conditional statements show that the reverse is happening: "When correctly fitted they are good. But because they are cheaper, people tend to overlight areas," or "If it is very red and very dim, these could reduce light pollution—they seem never to be red nor dim." The conditionality of positive observations resonates with Matthew Gandy's diagnosis: "The transition to LED technologies", said the geographer and urbanist, "illustrates a divergence between technologically infused environmental rhetoric and science-based ecological discourse, because the emerging political momentum to reduce light pollution is being offset by a new generation of energy-efficient technologies . . . " [37] (p. 1103).

## 4. Discussion: Sustainable LED Visions, Unsustainable Lighting Practices?

The expert statements presented in this study suggest that LED technology is not inherently good or bad, but is applied in contexts and used in ways that either contribute to or reduce light pollution and light-related nuisances. Design and installation-related challenges are at the root of the contradictory evaluation of LED technology. While bad choices lead to rebound effects, extremely bright white lighting and glare, good designs and installations offer a great opportunity for more sustainable, energy-efficient and eco-friendly lighting. This is reflected in two statements by the same light pollution expert. He argues that LEDs with color temperatures below 3000 Kelvin, fully shielded, with motion detection and "off when not needed or appropriate" can reduce light pollution, while LEDs above 3000K, unshielded as they are on the market now "make the night look like daytime" and contribute to light pollution.

Focusing on the relationship between LED technology and light pollution, it is striking to note that problems tend to be associated with the status quo of the innovation process, while the potential of LED lighting is described as an unrealized possibility (48.3% of the pro-LED statements are expressed in the conditional tense, compared to only 7.6% of the contra-statements). From a sociological innovation perspective, the problematization of the current state of affairs can be interpreted as a user response to "unsatisfactory innovation" [16]. In this regard, the lighting professionals and light pollution experts who participated in our project came to the same conclusion. Most of them agreed that better design and planning are key (Figure 4) and recommended educating lighting professionals and light users, so that they can design, install and use LEDs more appropriately [18] (pp. 199–204). At the same time, the description of less light polluting LED fixtures in the conditional can be understood as a critique of existing lighting practices set against a vision of more sustainable lighting designs and installations, which is shared by both light pollution experts and lighting professionals.

The experts' emphasis on design choices in the product development, planning and installation process highlights an important aspect: Innovative technological products alone will not solve the problem. In theory, customizable, controllable and color-changeable LED technology can fulfil the criteria of eco-friendly lighting. In practice however, innovations rarely encounter a tabula rasa. As social scientific research on technology and innovation shows, they are produced and adopted in specific socio-cultural contexts where existing infrastructures and institutions create path-dependencies and standards, and where users and their practices matter and shape technologies [38–41].

In the case of LEDs, unquestioned business as usual and lighting routines can diminish the potential for sustainable solutions. The prevailing positive connotations of illumination [42] and the sinking maintenance costs of lighting [43] create a favorable climate for rebound

effects [37,44]. Unquestioned industry standards that are designed to make products and lighting systems compatible with global production and usage [45] might not lead to the most sustainable project- and situation-specific LED applications. Instead, light specifiers, planners and users require professional skills and knowledge of both LED technology and application contexts, before they are capable of interpreting technical standards in a way that meets and mediates their specific needs [21,46].

In this situation, light pollution experts and ALAN research communities problematize ecologically unfriendly and unhealthy lighting practices and standards [47]. Their evidence on light pollution draws attention to the paradoxical situation and is already inspiring lighting professionals and decision makers to question and reconsider their professional practices [48–50].

There are also more tangible signs that concerns regarding the negative health and ecological effects of bright blue-rich LED lighting are being heard, and are affecting luminaire choices and lighting design. With regard to the application of LEDs, it seems that after a wave of cool-white LED installations, the color temperatures in street lighting are now getting lower. In 2017, the US Department of Energy (DOE) noted even a "small number of LED area/roadway luminaires are listed with a CCT of less than 2100 K. Though not common, this illustrates that the spectral power distribution of LED products can be tailored to meet desired goals." [51] (p. 7). With regard to LED product choices, policymakers have begun to promote a more responsible adoption of LED technology. To give an example, the 2019 EU Green Public Procurement Criteria for Road Lighting and Traffic Signals offers guidance on how to consider more than price and maintenance costs, and has newly introduced a lighting principle called ARLA—As Low As Reasonably Achievable [52]. John Barantine, Public Policy Director of the International Dark Sky Association (IDA), positively highlighted this mindfulness as "key to a global reduction in both light consumption and sky glow" [53]. Last but not least, powerful players within the lighting industry have begun to adapt their product portfolios, revise lighting standards and offer technical guidelines for adequate lighting [54]. Yet, design propositions for sustainable lighting from within the lighting field are founded on the sustainability of the lighting business. From this economic point of view, innovation is key, digital lighting control is more desirable than low-tech solutions, and PC amber LEDs are better than old-school high-pressure sodium lamps, even if it is debatable whether LEDs are more sustainable than HPS. In practice, LED installations do not always exploit the advantages of the innovation such as their directed light distribution. As a result, not all LED solutions are more energy efficient than HPS sources [55,56].

From a social-scientific perspective, different perspectives and different ways of producing and presenting evidence are linked to power asymmetries [57]. Well-established, institutionalized photometrical evidence that forms the epistemic basis for light engineering, light planning and lighting standards is therefore more likely to shape innovation and LED installations than the perspectives of light pollution experts who operate outside the lighting field and produce knowledge that might be at odds with road safety and good visibility.

Non-knowledge, also described as ignorance, is another issue [14]. Our findings suggest that the negative side effects of LED lighting can be quite easily avoided through well-informed product and design choices. Such sustainable design choices become more likely when builders and investors are willing to commission and pay lighting professionals in the first place. They should know how to avoid bad product choices, bad designs, glare and overlighting and be able to interpret standards [46]. As our interview partners pointed out, binding regulations can offer hard incentives to solicit such professional expertise in lighting projects [18] (pp. 123–168).

To conclude, the focus on paradoxes has the potential to challenge power-knowledge and ignorance. Paradoxes draw attention to incomprehensive or limited assumptions about actual contexts and empirical evidence that questions established lighting practices and principles. In this sense, the LED paradox can be understood as an invitation to consider new, surprising and inconvenient truths about what has been sold and described as a sustainable revolution in lighting [58]. It is also an invitation to acknowledge and meet complex, competing and potentially contradictory ecological, economic and social demands. Achieving such solutions with LED lighting can be challenging and raises issues that

are more socio-political than technological. The challenge is obvious when we look at contemporary lighting conflicts, where conservationists or astronomers negotiate or fight with municipalities and private actors over adequate light levels and color temperatures [36]. Boundaries between the ecological, economic and social dimensions of sustainable lighting are thereby not always clear: Energy efficiency is not only an environmental argument, but also an economical one [1]. Avoiding light pollution can be considered eco-friendly, but also economically relevant when night-time tourism is at stake [59]. Finally, public debates and negotiations over light in public spaces reveal that the social benefits of lighting are all but evident [60,61]. Against this backdrop, a specific LED solution might be the most sustainable in a given context, but that does not always translate into any general principle regarding a specific LED product, lighting system or usage.

## 5. Conclusions: The Dilemma Behind the LED Paradox

Since the 1960s, innovators and policy makers have promoted LED technology as the long sought-after key to sustainable lighting [62]. Today, it seems that the vision has partly become a reality, as LED luminaires are replacing conventional light sources worldwide. In terms of energy efficiency, LEDs have met or even exceeded the innovators' expectations [63]. The energy efficiency of the technology and the comparatively low maintenance requirements of long-lasting LED modules seem to justify the sustainability claim although there is an ongoing debate regarding the life span of LED technology and maintenance costs under different real-world conditions. Nevertheless, LED products are increasingly sold and used not only for indoor applications, but also for outdoor lighting [64]. Our survey results reflect this trend. Nearly 80% of the survey respondents stated that the introduction of LED outdoor lighting strongly affects their light-related activity (see Appendix A). However, the expert feedback also confirmed that LED technology can have paradoxical effects. While some argue that using the innovation "means more money in your pocket, and a healthier planet," [1] light pollution experts and an increasing number of lighting professionals are convinced that it really means more money in some pockets and an unhealthier planet. They also highlight that the search for the cheapest products and cost-efficient lighting systems rarely leads to the best solutions, but rather to rebound effects and an increasing loss of nighttime darkness on our planet [65,66]. In this situation, the expert participants in our project acknowledged both the potential and drawbacks of LED technology, and highlighted the role of expertise as a way to minimize unintended side effects and nasty surprises.

The expert feedback further suggests LED technology only offers a potential for designing lighting in more sustainable ways. The new technology is not just a product that is ready for use. On the contrary, as sociologists show, technologies are configurations of technical artifacts, larger technological systems such as digital networks or electricity supply grids, and users with specific skills and experiences. As such, they either work or fail [17] (p. 330). Therefore, innovative technologies can only fulfil their promise if their users live up to the novelty, learn how to use and apply new products and processes, and adapt or make them match existing socio-technical contexts to their full effect. In other words, sustainable lighting requires not only innovative products, but also innovative lighting practices. The new characteristics of LED technology challenge their producers and users to innovate, to test the technology in the real world and to update their lighting practices, in order to take full advantage of the new technology. At the same time, the light pollution debate around LEDs seems to have raised awareness about the unintended, negative side effects of artificial light at night, sometimes accompanied by a positive reevaluation of darkness [42,67,68]. This sociocultural innovation might offer a fresh and eco-friendly evaluative frame for LED lighting choices.

However, our project and this paper have only tapped into a specific aspect of sustainable LED lighting, neglecting many others. With regard to light pollution, we have focused on "ecological light pollution" and mentioned the potential health effects of "anthropological light pollution", but not the socio-economic effects of "astronomical" light pollution [20] (pp. 9–10). We have also ignored emerging concerns regarding the visual negative health effects of blue-rich LED lights and flicker [12] as well as issues with LEDs for indoor lighting [69–71]. There is also much more

to say about trade-offs between the economic, social and ecological aspects of lighting as opposed to the preservation of natural darkness [18] (pp. 123–168). Even more importantly, our focus on light pollution misses a number of crucial questions related to the sustainability of LED lifecycles. These questions concern the exploitation and recycling of raw materials and the sustainability of value and production chains [72–74].

To conclude, the LED paradox described in this paper suggests that sustainable lighting is not a technological artifact, but subject to negotiations. What counts as sustainable is dependent on the context and the audience. The question of how to achieve sustainable lighting is thus a question that innovators and users must explore together, in specific lighting projects and with a considerate eye for the environments that will be affected by the light, and which might be better off in darkness. The stakeholders in such negotiation processes possess neither the same knowledge, nor the same power to produce compelling evidence or legitimize their knowledge claims. There are also tradeoffs: The LED paradox draws attention to the fact that energy-efficient white LED light might be the least eco-friendly type of LED light, and in some cases even less socially acceptable. This dilemma is no longer paradoxical, but a challenge for light planning and lighting design.

**Supplementary Materials:** Background information on the project is available online at www.ufz.de/light-pollution.

**Author Contributions:** Conceptualization, N.S.-R., J.M. and E.D.; formal analysis, N.S.-R. and M.S.; funding acquisition, N.S.-R.; methodology, N.S.-R.; visualization, N.S.-R.; writing—original draft, N.S.-R.; writing—review and editing, J.M. and M.S.

**Funding:** This research received no external funding.

**Acknowledgments:** We would like to thank the anonymous reviewers for their valuable feedback. We are also very grateful to the many experts who participated in our "Global Discussion" on light pollution in 2018. The insights presented in this article are based on what we learnt from the participants of our online expert survey and our thematic expert discussions. We hope to continue this discussion in the future and beyond this article.

**Conflicts of Interest:** The authors declare no conflict of interest.

**Appendix A**

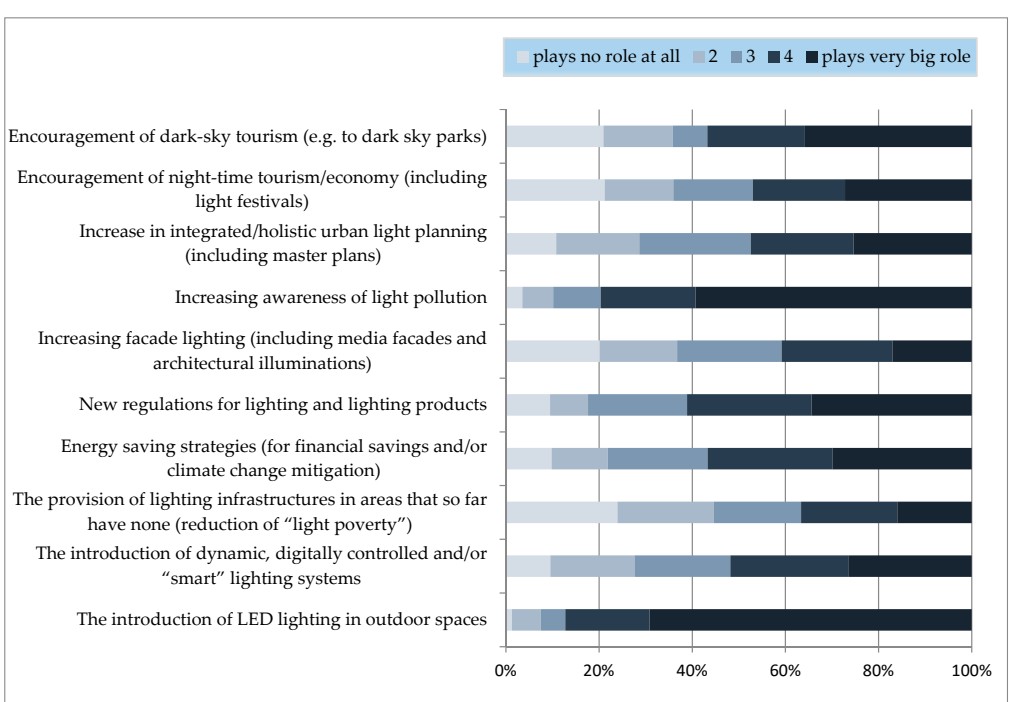

**Figure A1.** "To what extent do the following trends affect your light-related activitie(s)?."

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
