# Peer review of "The LED Paradox: How Light Pollution Challenges Experts to Reconsider Sustainable Lighting"

_sustainability, doi:10.3390/su11216160_

Round 1
Reviewer 1 Report
This paper present results from a survey of both light pollution experts and lighting professionals. The results are written in a positive light and this manuscript should be published to help move the lighting efforts of both light pollution experts and lighting professionals forward. I have very few minor comments.
153: This whole sentence doesn't make sense, please rewrite.
182: I am not sure "parcel" is the right word
251: play not plays
252: sixth not sixths
343: You need to start the sentence with a word, not the number. So "Seventy-nine percent"
405: This appendix could use more of a legend to explain that the actual statements mean. For example what does "Increase in" mean?
Acknowledgements!!!! FILL IT IN and perhaps thank all of the survey participants?!!!
Author Response
Dear reviewer, thank you very much for reading our paper so carefully! We have checked all your points (see below) and are pleased by your positive overall response.
153: This whole sentence doesn't make sense, please rewrite. > Done 182: I am not sure "parcel" is the right word > I changed the sentence to:“many of them consider excessively bright light levels and glare as effects of poor design and hance a result of problematic planning and installations."
251: play not plays >I think “plays” is correct: “The introduction… plays a role.” 252: sixth not sixths > I've changed it 343: You need to start the sentence with a word, not the number. So "Seventy-nine percent" > I've changed it to “Nearly 80% of the respondents…” for consistency reasons. 405: This appendix could use more of a legend to explain that the actual statements mean. For example what does "Increase in" mean? > Sorry, this was only a formatting problem. I changed the size of the table. Now you can read the sentences properly. Acknowledgements!!!! FILL IT IN and perhaps thank all of the survey participants?!!! > Of course! I've done it and also thank the reviewers…Best regards and thank you again!
Nona Schulte-Römer
Reviewer 2 Report
The discussion made in this manuscript is certainly interesting, because it develops around a theme (overall sustainability of LED lighting) for which the scientific community has not yet reached a definitive and shared position. For the purpose identified by the authors for this study, I found the manuscript clear and well structured, even if it does not have the classical structure of a research study where materials, methods and results are well highlighted.
Very interesting are the results of the survey through questionnaires.
The main obstacle to the publication of the manuscript in my opinion derives from the poor blibliographical references both in quantitative terms and, in some cases, qualitative terms, with respect to what such a study would require. Given that the authors, among the objectives of the study state "... We then present expert feedback on the LED dilemma with special attention to the positions and views of lighting professionals and light pollution experts. In our discussion, we reflect on these findings with regard to existing research on sustainable LED lighting.." this comparison with the scientific literature must be strengthened. The comparisons to be strengthened most in my opinion are those related to sustainability and interaction with human health of LED lighting. Here are some useful research for doing it.
https://doi.org/10.3390/su10103674
https://doi.org/10.3390/su8070618
10.1016/j.buildenv.2019.01.022
https://doi.org/10.3390/su71013454
https://doi.org/10.1109/EEEIC.2016.7555791
https://doi.org/10.2351/1.5118592
In the histogram shown in Appendix A, it should be possible to read the categories on the vertical axis completely.
Author Response
Dear reviewer, we thank you very much for taking the time and effort to read and comment on our article. We were pleased by your positive overall response and found your recommendations and advice very helpful.
Based on your feedback we have revised the manuscript in the following way:
"The discussion made in this manuscript is certainly interesting, because it develops around a theme (overall sustainability of LED lighting) for which the scientific community has not yet reached a definitive and shared position. For the purpose identified by the authors for this study, I found the manuscript clear and well structured, even if it does not have the classical structure of a research study where materials, methods and results are well highlighted.
Very interesting are the results of the survey through questionnaires."
Thank you!
"The main obstacle to the publication of the manuscript in my opinion derives from the poor blibliographical references both in quantitative terms and, in some cases, qualitative terms, with respect to what such a study would require."
We fully agree! In fact, we had been aware of these shortcomings when submitting the paper, but to meet the Special Issue deadline we postponed further improvements to the revision. We have now taken the chance to better explain how our findings relate to existing research in both social studies of science, technology and innovation (e.g. "paradoxes" 58-66, rewritten discussion line 395 onwards) as well as research on LEDs and light pollution (e.g. lines 216-216, but also other parts). We've also added more information from our expert discussions on the perception of lighting and how that affects color temperature choices (e.g. 227-251). We thank you very much for the paper suggestions (the links you provided, see below)!
"Given that the authors, among the objectives of the study state "... We then present expert feedback on the LED dilemma with special attention to the positions and views of lighting professionals and light pollution experts. In our discussion, we reflect on these findings with regard to existing research on sustainable LED lighting.." this comparison with the scientific literature must be strengthened."
Yes, absolutely. To solve the problem, we did a bit more than minor revisions. In particular, we made the following changes and additions:
Throughout the paper, we have spelled out more clearly that we are interested in the complexity and unresolved questions of sustainable lighting. In particular, non-knowledge, knowledge asymmetries and context specific demands call for societal negotiations across expert perspectives. (See for instance the part on standardization in the discussion and the added material on the color temperature dilemma, 227-251).
Abstract (last sentence rewritten): "This expert feedback shows that ‘sustainable LED lighting’ goes far beyond energy efficiency as it raises complex design issues that imply stakeholder negotiation. It also suggests that the LED paradox may be solved in context, but hardly in principle."
Last but not least, the contribution of the paper to technical issues and positive knowledge about sustainable lighting is marked by limitations. We have made that clearer in the conclusion (497-505).
"The comparisons to be strengthened most in my opinion are those related to sustainability and interaction with human health of LED lighting. Here are some useful research for doing it."
We see your point, but our research does not offer exciting results in this respect. Our data shows that design is crucial, but it does not offer new insights about how LEDs negatively affect human health or not. Nevertheless, we added information and references on that point (216-226).
We added the following links you offered to our references:
https://doi.org/10.3390/su10103674 (added, see conclusion) https://doi.org/10.3390/su8070618 (indoor and sport facilities, not light pollution) 10.1016/j.buildenv.2019.01.022 (link did not work) https://doi.org/10.3390/su71013454 (added, see conclusion, ref. 67) https://doi.org/10.1109/EEEIC.2016.7555791 (added, ref. 27)https://doi.org/10.2351/1.5118592 (added, see conclusion, ref. 69)
"In the histogram shown in Appendix A, it should be possible to read the categories on the vertical axis completely."
Of course, this was only a formating mistake. Thanks for pointing it out!
One last point, we found a mistake in our calculations of overlaps and have corrected Figure 1 and 2 accordingly.
Reviewer 3 Report
Dear authors,
I'm glad that you discuss this topic and is a very important one but the title can be misleading as this is is not a review paper about the LED ecological impacts as the title cloud represent. I suggest you to change the title so represent more the real content of the article that is the discussion on the points of view of the lighting community.
A title like that should not only include the environmental impact of LEDs in light pollution, but also extensive analysis of the quantitative impacts of the LEDs in different scenarios and impacts. From the light pollution in different color temperatures versus other light sources, life cycle assessments, CO2 emissions, waste, etc.
Here two references that could help you on this topic.
Tähkämö, L., Räsänen, R. S., & Halonen, L. (2016). Life cycle cost comparison of high-pressure sodium and light-emitting diode luminaires in street lighting. The International Journal of Life Cycle Assessment, 21(2), 137-145.
Tähkämö, L., & Halonen, L. (2015). Life cycle assessment of road lighting luminaires–comparison of light-emitting diode and high-pressure sodium technologies. Journal of Cleaner Production, 93, 234-242.
Alternatively, you could change the title to a more realistic representation of what the article has that is a compilation of the opinions on the light pollution community and one proposal for a solution.
Some of the claims of the article are not been supported with data. or the data goes in the opposite direction of what the authors claim. The authors should not be biased by the manufacture's propaganda, and have to provide data. Example: LED's have "The energy efficiency of the technology and the comparatively low maintenance requirements of long-lasting LED modules justify the sustainability claim."
It is just a claim, the authors have data available, like Revision of the EU Green Public Procurement Criteria for Road Lighting and traffic signals, Annex IV. Examples of Life Cycle Costing. The maintenance costs of LEDs are higher than the HPS.
Also, life of LEDs, is highly discussable.
https://eu.detroitnews.com/story/news/local/detroit-city/2019/05/07/detroits-led-streetlights-going-dark-after-few-years/3650465002/
https://www.tandfonline.com/doi/full/10.1080/10643389.2017.1370989
Also efficiency... as is also discussed on the GPP road lighting, where low-pressure sodium lights where not included and they are way more efficient that all LEDs.
Although there is some papers cited that are related to quantitative impact of the LEDs, many of the relevant ones are missing.
Aubé, M., Roby, J., & Kocifaj, M. (2013). Evaluating potential spectral impacts of various artificial lights on melatonin suppression, photosynthesis, and star visibility. PloS one, 8(7), e67798.
Falchi, F., Cinzano, P., Duriscoe, D., Kyba, C. C., Elvidge, C. D., Baugh, K., ... & Furgoni, R. (2016). The new world atlas of artificial night sky brightness. Science advances, 2(6), e1600377.
Luginbuhl, C. B., Boley, P. A., & Davis, D. R. (2014). The impact of light source spectral power distribution on sky glow. Journal of Quantitative Spectroscopy and Radiative Transfer, 139, 21-26.
Sánchez de Miguel, A., Aubé, M., Zamorano, J., Kocifaj, M., Roby, J., & Tapia, C. (2017). Sky Quality Meter measurements in a colour-changing world. Monthly Notices of the Royal Astronomical Society, 467(3), 2966-2979.
Jägerbrand, A. (2015). New framework of sustainable indicators for outdoor LED (light emitting diodes) lighting and SSL (solid state lighting). Sustainability, 7(1), 1028-1063.
Kinzey, B. R., Perrin, T. E., Miller, N. J., Kocifaj, M., Aube, M., & Lamphar, H. A. (2017). An investigation of LED street lighting's impact on sky glow (No. PNNL-26411). Pacific Northwest National Lab.(PNNL), Richland, WA (United States).
This are papers that mainly support the LEDs that produce more environmental impacts. Sure you will easily find non peer-reviewed articles that claim the opposite like this:
https://www.earthtronics.com/leds-and-climate-change/
The challenge here is to find peer-review articles. GPP might be considered as peer reviewed as there is a discussion with the stakeholders, but not the DOE documents.
Here is one, but have serious mistakes, and the article recognizes that the amber light is the best for the environment.
Peña-García, A., & SÄ™dziwy, A. (2019). Optimizing Lighting of Rural Roads and Protected Areas with White Light: A Compromise among Light Pollution, Energy Savings, and Visibility. LEUKOS, 1-10.
if the authors do not find enough of them, maybe the title should change to a more aggressive one. Maybe "LED ilusion" to be soft.
In general, I agree with the statement of the authors "To conclude, the LED paradox suggests that sustainable lighting is not a technological artifact, but subject to negotiations." In terms that is negotiable which environmental variables we should prioritize and which lighting scenarios we will consider. But, more quantitative and integral environmental impact assessments are needed it because of the complexity and the propaganda and financial pressure.
Overall, the article is great. My main concern is that the title is not accurate and some of these things are missing.
Author Response
Dear reviewer, we thank you very much for taking the time and effort to read and comment on our article. We were pleased by your positive overall response and found your recommendations and advice very helpful!
Based on your feedback we have revised the manuscript in the following way:
"I'm glad that you discuss this topic and is a very important one but the title can be misleading as this is is not a review paper about the LED ecological impacts as the title cloud represent. I suggest you to change the title so represent more the real content of the article that is the discussion on the points of view of the lighting community."
Very good point! Thank you, we have changed the title.
"A title like that should not only include the environmental impact of LEDs in light pollution, but also extensive analysis of the quantitative impacts of the LEDs in different scenarios and impacts. From the light pollution in different color temperatures versus other light sources, life cycle assessments, CO2 emissions, waste, etc. ..."Alternatively, you could change the title to a more realistic representation of what the article has that is a compilation of the opinions on the light pollution community and one proposal for a solution."
Absolutely right. We have taken your advice by a) changing the title. We have also spelled out more clearly throughout the paper that we are interested in the complexity and unresolved questions of sustainable lighting. In particular, non-knowledge, knowledge asymmetries and context specific demands call for societal negotiations across expert perspectives. (See for instance the part on standardization in the discussion and the added material on the color temperature dilemma, 227-251). See also our additional explanations regarding our perspective on the "paradox" (introduction 85-66).
b) In the conclusion, we have made clear that this paper does not offer insights into other LED impacts and evaluations in the context of sustainable lighting. We have therefore spelled out the limitations of the paper more clearly and added references to live cycle assessment, health effects and the issue HPS vs. LED. Thank you for your literature recommendation in this respect (Tähkämö and Halonen...)
Some of the claims of the article are not been supported with data. or the data goes in the opposite direction of what the authors claim. The authors should not be biased by the manufacture's propaganda, and have to provide data. Example: LED's have "The energy efficiency of the technology and the comparatively low maintenance requirements of long-lasting LED modules justify the sustainability claim."
Thank you very much for this important comment. We fully agree and do not want to write as if we uncritically 'swallow' PR messages. We have changed the formulation of that sentence to
"The energy efficiency of the technology and the comparatively low maintenance requirements of long-lasting LED modules seem to justify the sustainability claim although there is an ongoing debate regarding the life span of LED technology and maintenance costs under different real-world conditions."(469-472).
We have also added revised the papers you recommended, added the papers that related to our findings and also added numerous other references.
However, I did not find enough scientific evidence that supports the opposite claim -that LEDs are worse than HPS, etc. Instead, it seems that it all comes down to considerate design questions. If LEDs are installed in an inconsiderate and irresponsible way they are worse than HPS, but they can do better... (see also Tähkämö and Halonen...)
This is why we have not changed the title to LED illusion but stick with the paradox.
We thank you very much for your comments and suggestions, also for your very interesting points regarding peer review and DOE reports (I very much appreciate this information and will keep it in mind). I hope you agree that our changes and additions to the paper (including restructuring of the discussion and conclusion) have improved the presentation of our findings and our argument.
Thanks again and best regards,
Nona Schulte-Römer (on behalf of the team).